# Australian Women’s Experiences of Establishing Breastfeeding after Caesarean Birth

**DOI:** 10.3390/ijerph21030296

**Published:** 2024-03-03

**Authors:** Sharon L. Perrella, Sarah G. Abelha, Philip Vlaskovsky, Jacki L. McEachran, Stuart A. Prosser, Donna T. Geddes

**Affiliations:** 1School of Molecular Sciences, The University of Western Australia, Crawley, WA 6009, Australia; sarah.abelha@uwa.edu.au (S.G.A.); jacki.mceachran@uwa.edu.au (J.L.M.); stuart@westernobs.com.au (S.A.P.); donna.geddes@uwa.edu.au (D.T.G.); 2ABREAST Network, Perth, WA 6000, Australia; 3UWA Centre for Human Lactation Research and Translation, Crawley, WA 6009, Australia; 4Western Obstetrics, Balcatta, WA 6021, Australia; 5Department of Mathematics and Statistics, School of Physics, Mathematics and Computing, The University of Western Australia, Crawley, WA 6009, Australia; philip.vlaskovsky@uwa.edu.au

**Keywords:** breastfeeding, lactation, birth, caesarean, postpartum, midwifery

## Abstract

Breastfeeding exclusivity and duration rates are lower after caesarean birth, yet the factors contributing to these are not well understood. This mixed-methods study used an anonymous online questionnaire to examine the facilitators and barriers to establishing breastfeeding as identified by Australian women after a caesarean birth. Quantitative data were reported using descriptive statistics, and multivariable models were used to determine the factors associated with breastfeeding outcomes including the timing of breastfeeding initiation, birth experience, and commercial infant formula use. Qualitative data were analysed using an inductive thematic analysis. Data were obtained for N = 961 women, of which <50% reported skin-to-skin contact during breastfeeding initiation. The barriers to breastfeeding included aspects of clinical care and reduced mobility, while unrushed care, partner support, and physical help with picking up the baby were helpful. Following a non-elective caesarean birth, women had half the odds of early breastfeeding initiation (OR = 0.50; 95% CI: 0.36, 0.68; *p* ≤ 0.001) and 10 times the odds to report a negative birth experience (OR = 10.2; 95% CI: 6.88, 15.43; *p* < 0.001). Commercial milk formula use was higher in primiparous women (OR = 2.16; 95% CI: 1.60, 2.91; *p* < 0.001) and in those that birthed in a private hospital (OR = 1.67; 95% CI: 1.25, 2.32; *p* = 0.001). Pain and reduced mobility, as well as conflicting and rushed care, negatively impacted breastfeeding after a caesarean birth, while delayed breastfeeding initiation, higher pain ratings, and negative birth experiences were more common for women that birthed by non-elective caesarean. This study adds valuable insights into the physical, emotional, and clinical care needs of women in establishing breastfeeding after a surgical birth. Clinical staffing and care should be modified to include full access to partner support to meet the specific needs of breastfeeding women after a caesarean birth.

## 1. Introduction

Birth by caesarean section (CS) involves major abdominal surgery that is associated with pain, reduced mobility, and maternal and infant health complications [1,2]. Furthermore, breastfeeding outcomes may be compromised after CS birth, with evidence of delayed initiation, a higher incidence of breastfeeding difficulties, and shorter breastfeeding exclusivity and duration when compared to vaginal birth [3,4]. The association with shorter breastfeeding duration has been reported in several countries and may be more marked after an elective CS birth [5]. Indeed, a Canadian cohort study found that, compared to vaginal birth, women birthing by elective CS were 61% more likely to have stopped breastfeeding by 12 weeks postpartum [6]. It is difficult to further quantify these differences as breastfeeding duration data and associated birth modes are not consistently collected and reported [7].

The early initiation of breastfeeding, that is, within an hour of birth, has important maternal and infant health benefits, with evidence of positive associations with breastfeeding exclusivity and duration [5]. However, the early initiation of breastfeeding is significantly less likely after a CS birth compared to vaginal birth due to barriers including postoperative pain and limited skin-to-skin contact [8,9].

While the WHO recommends feeding infants only breast milk in the first six months with continued breastfeeding to two years and beyond, both breastfeeding exclusivity and duration rates are lower for infants born by CS birth compared to those born by vaginal birth [8,10]. Exclusive breastfeeding during the postnatal hospital stay is positively associated with breastfeeding duration, and as infants born by CS are at increased risk of being fed formula in the days after birth, this may, in part, explain their shorter breastfeeding duration [11,12]. Maternal psychological distress is more prevalent after a CS birth and has been associated with suboptimal breastfeeding outcomes [13,14]. Furthermore, early postnatal breastfeeding difficulties are reported to be more common after an unplanned CS birth [3,6].

Dyads that initiate breastfeeding after a CS birth face greater challenges and have shorter breastfeeding durations compared to those after a vaginal birth. While the results from a few studies indicate higher rates of early breastfeeding difficulties and shorter breastfeeding durations for dyads after an elective CS [3,4,6], further research is needed to explore the differences in early breastfeeding experiences according to the CS birth mode. Socio-demographic and psychosocial factors, such as maternal educational attainment and infant feeding attitudes, have been identified as influencers of infant feeding for women in high-income countries [10,15]. However, there is sparse contemporary data that reflect maternally identified barriers and facilitators to the establishment of breastfeeding, which can inform clinical practice. The rates of CS births are high, with the global and Australian rates currently being 21% and 34% [16]. Therefore, it is imperative that the unique breastfeeding challenges and support needs of women after a CS birth are anticipated and met. The exploration of women’s experiences of establishing breastfeeding after a CS birth will inform clinical care on how to improve breastfeeding outcomes and subsequent maternal and infant health. The aim of this study was to determine Australian women’s experiences and the facilitators and barriers to the establishment of breastfeeding after a CS birth.

## 2. Materials and Methods

A mixed-methods descriptive study design was employed using an anonymous online questionnaire on the experience of establishing breastfeeding in hospital and in the first two weeks at home after a CS birth. This paper reports women’s experiences of breastfeeding during their hospital stay. The study design enabled descriptions of participants’ demographic, birth, and postpartum characteristics to be obtained while also capturing the maternal voice, thereby facilitating an understanding of women’s varied experiences that is vital in informing healthcare policy and practice [17].

Participants were recruited via social media posts on Instagram and Facebook that target Australian women of childbearing age. Inclusion criteria were women that gave birth by CS to a term infant (born 37–42 weeks gestation), lived in Australia, were able to read English, and ≥18 years of age. Women that had given birth to multiple birth infants were not specifically excluded. To minimise the potential for recall bias, responses from women that gave birth by CS > 12 months ago were excluded.

Purposive sampling was used to capture a balanced representation of women’s experiences with data saturation of subgroups according to CS birth classification and parity. Caesarean birth may be broadly classified as elective, i.e., planned before the onset of labour or obstetric complications (elective lower uterine segment caesarean section, ELUSCS), or unplanned/emergency due to obstetric complications (non-elective lower uterine segment caesarean section, NELUSCS) [18]. As NELUSCS births are more likely to be associated with maternal stress depending on the degree of concern for maternal and/or foetal life or health, women’s experiences of establishing breastfeeding may differ based on the CS birth type. Similarly, the experiences of women may differ based on parity. Therefore, we aimed to recruit *n* = 300 women, with *n* = 50 in each subgroup according to parity and CS birth type, i.e., primiparous ELUSCS and NELUSCS, multiparous, ELUSCS after previous CS, ELUSCS after previous vaginal birth, NELUSCS after previous CS, and NELUSCS after previous vaginal birth.

Participants completed an online 67-item questionnaire that took approximately 20 min to complete. The questionnaire was developed with input from consumers and experienced clinician researchers to ensure internal validity. External validity was increased by using broad inclusion criteria to achieve a representative study sample. Reliability was not tested as the reliability of maternal recall of birth and breastfeeding events are moderate and high, respectively, at 12 months postpartum [19,20]. The rigor of the qualitative data analysis and interpretation were achieved through the implementation of a recognised methodology as described by Braun and Clarke [21]. The questions captured demographic data and addressed maternal experiences and factors influencing breastfeeding establishment. Details of birth satisfaction, skin-to-skin contact, mobility, pain, and infant feeding methods were recorded. Likert scale items were used to determine maternal satisfaction with different elements of the breastfeeding experience, including postpartum care and support. Participants were invited to respond to open-ended questions regarding perceived early postnatal barriers and facilitators to breastfeeding, with no word limit set for qualitative responses.

Qualitative data were analysed by a subgroup using an inductive thematic analysis conducted as per Braun and Clarke’s framework [21]. Specifically, two researchers (SGA and SLP) independently read over the responses many times to ensure they were familiar with the data and methodically noted initial ideas, applied code words or phrases to identify the meaning of each response, and then collated the responses for each code. SA and SP then met to compare analyses and resolved any coding discrepancies through discussion until a mutual agreement was attained. Examination and comparison of analyses by two researchers reduced bias and supported the validation of the codes. When no new codes emerged, i.e., saturation of the data was achieved, the generated codes were interpreted, and common codes collated into themes that were further refined into subthemes. The investigators reviewed the subthemes to ensure agreement that all themes were adequately defined and described. The final qualitative data analyses involved the identification, selection, and explicit analysis of pertinent quotes for inclusion in the reporting of results.

Quantitative data were described using mean ± standard deviation (SD) for continuous variables and n (%) for categorical data. Univariable and multivariable logistic regression analyses were used to evaluate relationships between the early initiation of breastfeeding (i.e., within an hour of birth) and predictor variables of CS birth type, parity, and pain rating; commercial milk formula use in hospital and predictor variables of CS birth type, parity, pain rating, hospital setting (private or public), incidence of birth complications, and whether birth expectations were met; and negative birth experience (defined as ‘a little traumatic’ or ‘very traumatic’ AND ‘birth expectations were not met’) and CS birth type, parity, pain rating, and incidence of birth complications. Odds ratios and confidence intervals were reported. All analyses were performed using R Statistical Software (v4.3.1; R Core Team 2023), and statistical significance was set at a *p*-value of <0.05.

Study data were collected and managed using REDCap, a secure web-based software platform designed to support data capture for research studies [22]. Ethics approval was granted by the Human Research Ethics Committee of the University of Western Australia (2022/ET000174). Participants provided informed digital consent and were aware that their responses were anonymous and that they could exit the study at any time, for any reason.

## 3. Results

Women were recruited from May 2022 to August 2022 when *n* = 50 was achieved for all subgroups, resulting in over-recruitment for most subgroups while waiting to complete recruitment to ‘multiparous NELUSCS after previous vaginal birth’. Of the *n* = 1239 responses, *n* = 961 met the study criteria, with one subgroup having *n* < 50 (Figure 1).

The participants were 33.0 ± 6.2 years old and gave birth at 39.1 ± 1.0 weeks’ gestation; most (n = 646, 67.2%) held a tertiary qualification and were partnered (*n* = 901, 93.8%). The intended breastfeeding duration was 12.5 ± 6.4 months, with the women being at 6.1 ± 4.3 months postpartum at the time the survey was completed. Further participant characteristics are reported in Table 1.

### 3.1. Quantitative Findings

Almost half of the women (*n* = 441, 46.4%) first held their newborn infants for <15 min, and *n* = 430 (44.7%) reported skin-to-skin contact during the first breastfeeding attempt. Most reported early breastfeeding initiation (Table 2), with primiparous women being more likely to have a midwife provide physical assistance (*n* = 312, 65.0% vs. multiparous *n* = 186, 38.7%, *p* < 0.001). The CS birth type, parity, and maternal pain were all significantly associated with early breastfeeding initiation in a univariable analysis, but because the factors are somewhat confounded, we report the multivariable model that incorporates all three together (Table 3). Women that had an NELUSCS birth had half of the odds to breastfeed within an hour of birth compared to those who had an ELUSCS birth (OR = 0.50; 95% CI: 0.36, 0.68; *p* ≤ 0.001). Primiparous women had 32% lower odds to breastfeed within an hour of birth compared to multiparous women (OR = 0.68; 95% CI: 0.50, 0.94; *p* = 0.018). Finally, after adjusting for CS birth type and parity, there was no association with maternal pain rating (*p* = 0.13).

Compared to women that had an ELUSCS birth, those that had an NELUSCS birth reported higher pain ratings in the hours and days after birth (Table 2 and Table 3) and more frequently reported that pain interfered with breastfeeding (Table 3).

Overall, commercial formula milk or ‘formula’ use was reported by *n* = 308 (32.1%) (Table 4). An NELUSCS birth, primiparity, higher pain ratings, a private hospital setting, and birth complications were all significantly associated with formula feeding during the hospital stay in a univariable analysis (Table 5). However, as these factors are confounded, we report the multivariable model. The odds for reporting formula feeding were 2.16 (95% CI 1.60–2.91) times higher in primiparous women, 2.25 (95% CI 1.64–3.10) times higher for women that experienced birth complications, and 1.67 (95% CI 1.25–2.22) times higher for women that birthed in a private hospital (Table 5). Approximately one-third of women (*n* = 300, 31.2%) reported that it was easy to pick up the baby to breastfeed. In relation to care during the hospital stay, *n* = 611 (63.6%) women were satisfied with the breastfeeding support provided, and nearly half (*n* = 460, 47.9%) reported that they received conflicting breastfeeding information from staff.

While most women described their births as ‘quite easy’ or ‘difficult but overall okay’, reports of a negative birth experience, i.e., birth rated as ‘a little traumatic’ or ‘very traumatic’, together with unmet birth expectations, were higher for NELUSCS births (primiparous *n* = 158, 52%; multiparous *n* = 61, 55%) than for ELUSCS births (primiparous *n* = 16, *n* = 9%; multiparous *n* = 33, *n* = 9%). The CS birth type, parity, pain rating, and incidence of birth complications were associated with a negative birth experience in a univariable analysis. As the factors are somewhat confounded, we report the multivariable model that incorporates all four together. Women that had an NELUSCS birth had 10.2 (95% CI 6.88–15.43) times the odds to report a negative birth experience compared to those who had an ELUSCS birth. The odds to report a negative birth experience were 3.95 (95% CI 2.74–5.72) for women that had a birth complication, and 1.23 (95% CI 1.13–1.34) for those with a higher pain rating (Table 6).

### 3.2. Qualitative Findings

Qualitative data were provided by *n* = 767 women, of which *n* = 375 (49%) were primiparous. An inductive thematic analysis identified three main themes that characterised women’s experiences of establishing breastfeeding after a CS, including helpful and unhelpful aspects of care. The first theme, ‘experience of care’, comprised eight opposing subthemes, while the other two themes comprised interrelated subthemes (Figure 2). Direct quotes are integrated throughout the findings to both elucidate and provide evidence for the themes and subthemes. Codes are used to indicate each respondent’s parity and birth type, i.e., [P] = primiparous, [M] = multiparous, [ELUSCS] = elective lower uterine segment caesarean section, and [NELUSCS] = non-elective lower uterine segment caesarean section. For example, (M, NELUSCS) indicates the response of a multiparous woman after a non-elective CS birth.

#### 3.2.1. Theme I. Experience of Care

Qualitative data on the experience of care were received from *n* = 673 women of which *n* = 364 (54%) were primiparous. Various elements of care were fundamental to the experience of establishing breastfeeding, with some dichotomous subthemes identified (Figure 2). The dichotomous subthemes had predominant themes expressed in the majority of responses, while the minor themes with opposing views were expressed by few.

‘Midwives doing, not showing’ versus ‘helpful hands-on approach’.

Reflecting on their hospital stays, many women found physical assistance with breastfeeding and the absence of explanations to be unhelpful and even forceful. ‘*Wanted baby to find own way to nipple with support but nurse came in and forced baby onto nipple in a ruff [sic] manner without parental care*’ (M, ELUSCS). ‘*The lactation consultant would thrust the baby onto the boob, check the baby was making suck motions, tell you “your [sic] now breastfeeding” and leave. There was no explanation of how to latch etc*.’ (M, ELUSCS).

A few primiparous women appreciated a hands-on approach, including midwives who would ‘*come and latch him for me every time*’ (P, ELUSCS). For these women, physical help was typically not perceived as rough or forceful, although there were exceptions: ‘…*the midwife who essentially grabbed my child’s head like a softball and jammed my breast into her mouth got the best latch out of anyone*.’ (P, NELUSCS).

‘Too busy to help’ versus ‘midwife took the time’.

A predominant theme across all groups was staff busyness and rushed care as barriers to establishing breastfeeding, with perceptions that clinical staff were unable to spend adequate time helping with breastfeeding, and sometimes lacked patience as a result: ‘…*lack of help by staff as they were clearly short staffed & didn’t have much patience to help me for very long*’ (P, ELUSCS). “*It was painful to pick my baby up from the bassinet to be able to feed her…overworked staff had too many patients to see and limited time to help with breastfeeding or lifting baby out of bassinet and passing her to me*” (P, NELUSCS).

Some women avoided asking for help as they did not want to further burden the midwives: ‘*I was hesitant to seek assistance with breastfeeding during the night as I didn’t want to disturb midwife’s [sic] as I think they were quite busy’* (P, NELUSCS).

Whilst many women described a lack of adequate staff support, some felt overwhelmed by the attention of the clinical staff, with one woman identifying ‘*space from staff and visitors*’ (P, NELUSCS) as a facilitator to breastfeeding, and others reporting the importance of privacy, ‘*A private room so I could have my boobs out all the time*’ (P, NELUSCS).

Women across all groups highlighted the positive impact of an individual midwife that provided time and respectful care during their hospital stays: ‘…*one midwife who was kind, gentle and took the time to explain things to both me and my husband*’ (P, ELUSCS); ‘…*one midwife… took the time to sit with me in the nursery and hand express*’ (P, NELUSCS).

‘Conflicting advice confused me’ versus ‘varied clinical advice facilitated my learning’.

Although most women described the midwives’ breastfeeding support as helpful, many reported that the conflicting advice was unhelpful, disruptive, and confusing.

‘*… each midwife had their different ideas of how it should be done which was sometimes frustrating and not consistent’* (P, ELUSCS).

‘*Staff telling me to do different things. I was trying something one midwife suggested, only to be told I was doing it wrong by another midwife. It was confusing. I just wanted to find my groove with my baby, and I would ask if I needed help. But the staff loved to interfere’* (P, ELUSCS).

In contrast, a few women perceived the varied breastfeeding advice to be helpful as it provided a range of strategies to trial. One woman described a facilitator of breastfeeding to be ‘*having different midwife’s [sic]/lactation consultants offer and show me different ways of breastfeeding. I was then able to try all different ways myself and [this] has become useful since being home*’ (P, NELUSCS).

‘Experienced mums need help too’ versus ‘happy to be left alone’.

Some multiparous women voiced feeling neglected by midwives due to a perceived assumption that experienced mothers did not need breastfeeding support. One mother recalled, ‘*as a third time mum I felt like I was left to my own devices and I felt a bit brushed off until I asserted myself*’ (M, ELUSCS). This was particularly hard for women with previous breastfeeding difficulties, as one reported ‘*being a third time mum they assumed I knew what I was doing but I hadn’t successfully breastfed before*’ (M, ELUSCS), while others appreciated the support: ‘…*midwives making me feel I could ask questions even though I was a second time mother*’ (M, NELUSCS).

Some women mentioned their previous breastfeeding experience was helpful in establishing breastfeeding this time around. One woman reported ‘*I was confident to try different holds that I had learnt myself prior*’ (M, NELUSCS), while another was happy to be left alone: ‘*I felt more comfortable feeding and didn’t ask the midwives for a lot of advice (as it more often than not is conflicting) and just did my own thing*’ (M, NELUSCS).

#### 3.2.2. Theme II: Expectations

Qualitative data relating to maternal expectations were received from *n* = 264 women of which *n* = 200 (76%) were primiparous. The expectations of establishing breastfeeding were dichotomous amongst women; some perceived breastfeeding as a process of trial and error that may not always go smoothly, while others felt overcome by pressure and stress when breastfeeding issues were encountered.

All too much

Some women described the experience of establishing breastfeeding in hospital as being characterised by ‘*stress and anxiety*’ (M, ELUSCS). One woman reporting feeling ‘*pressure that baby wasn’t getting enough*’ (P, ELUSCS). For some, this was compounded by exhaustion, conflicting advice, and inadequate support, which ultimately led to breastfeeding cessation: ‘*The recovery of a major surgery completely got in the way of establishing good breastfeeding and there was little to no help from the hospital staff, I requested a lactation consultant but the hospital was too busy and I never saw one, I ultimately decided that the stress of not being able to establish a proper latch and the toll it took on my mental health was too much and I decided to formula feed*’ (P, ELUSCS). The strategies used to reduce the stress around breastfeeding included ‘*having a “try again in a bit” attitude to reduce pressure*’ (P, NELUSCS) and placing ‘*no pressure on* [myself]’ (M, ELUSCS).

Breastfeeding isn’t always easy, but it will work out

In contrast, some women expected that breastfeeding can be challenging and trusted that it could be achieved: ‘I believed in my ability to learn to breastfeed and didn’t expect it to be easy or perfect straight away, so I had a realistic expectation so when things didn’t go smoothly at first, I was not discouraged’ (P, NELUSCS). Women who held these expectations typically also expressed motivation to breastfeed: ‘I knew it was how I wanted to feed my baby. I knew it was normal for it not to be easy and trusting we would get there’ (P, NELUSCS).

#### 3.2.3. Theme III: Unable to Pick Up Baby

Qualitative data were received from *n* = 719 women of which *n* = 392 (55%) were primiparous. A commonly identified barrier to breastfeeding was difficulty in picking up the baby to breastfeed as a result of reduced mobility, physical barriers, and/or separation from the baby.

Reduced mobility: a multifaceted barrier

The effects of anaesthesia, postoperative pain, and restrictive medical devices interfered with the women’s mobility and abilities to pick up and position their babies at the breast. The women’s descriptions of these factors included ‘*the soreness of my wound when I get up to pick up baby*’ (P, NELUSCS), ‘*the bassinet was too high for me to reach her. In the beginning as I was paralysed and in pain from the chest down*’, (P, NELUSCS) and ‘*Not being able to pick up my baby myself due to the c section pain as well as blood pressure machine and IV drip attached to me*’ (P, NELUSCS).

Many women reported complete dependence on the availability of other people to access their baby for feeding: ‘it was difficult after surgery to get up and hold my baby and find a comfortable position. The ward was busy, so someone was not always available’ (P, NELUSCS).

Having their partners present during the hospital stay positively influenced the women’s experiences. Women that birthed in a private hospital, where ‘visiting hours’ are not applied to partners, identified their partners’ availability as being imperative to establishing breastfeeding: ‘*Having my partner be able to stay with me in hospital was the most helpful as he would pick up the baby and hand her to me every time she needed to feed*’ (M, NELUSCS). In contrast, when their partners were not allowed to stay overnight, many women described feeling alone and unsupported.

‘*After a traumatic birth in a public hospital, my partner was not allowed to stay overnight. I found it exceptionally traumatizing to be left several hours after major abdominal surgery, unable to pluck my baby from his crib and manage him on my own. I was in agony from the caesarean and I didn’t know how to breastfeed’* (P, NELUSCS).

‘*My husband was not allowed to stay outside of visiting hours. This coupled with the midwifes [sic] being very busy at night left me feeling like I had no support… The main motivation for me leaving the hospital so soon after birth was the fact I would have help from my husband at home’* (P, NELUSCS).

A woman who was additionally vulnerable as someone who spoke English as a second language poignantly illustrated the challenges of being left alone at night after a surgical birth:

‘*I was in the public hospital so I couldn’t get my husband to stay and help me with the baby overnight … that was absolutely complicate and really bad experience I never felt more lonely in pain, no available to move by myself … I need help for stand up and look after the baby, nurses no was all the time there … so much easier if I have my husband in there helping me, was hard to hold the baby, stand up with a c section to clean the baby many times overnight try to teach him to eat from my breast. Was absolutely traumatic and sad be by myself that 2 nights without husband…because the sistem rules he can’t stay to help so it affect my breastfeeding I think and I stop producing milk because was under stress’* (P, NELUSCS).

Separation from baby

Women who were separated from their newborns due to complications requiring the neonatal unit or adult special care admission described feelings of isolation. Regardless of the reason for separation, the women described a lack of care that left them feeling alone: ‘*nobody assisted me or I felt pressured to walk to the neonatal unit alone to feed*’ (P, NELUSCS). The women voiced feelings of neglect and inadequate support during their infants’ neonatal unit admissions: ‘*Midwives offered little to no support, (some) told me in the NICU I was irresponsible and wasting their time by refusing formula and asking to be called into the NICU for every feed. They gave him formula once and I did not agree to this I was incredibly hurt and felt disrespected as his growth was perfectly normal*’ (P, NELUSCS).

‘*I spent the first night in the ICU following a postpartum haemorrhage as they did not think they could monitor me adequately on the maternity ward. They said they would bring my baby to me regularly for feeding, but they were busy and didn’t…’* (P, NELUSCS).

## 4. Discussion

In this study, most of the women initiated breastfeeding within an hour of the CS birth while making skin-to-skin contact with their infant. However, reduced mobility and pain hindered their subsequent breastfeeding experiences, with many women reporting difficulty in picking up their babies to breastfeed. The physical help and support of a partner during the hospital stay was considered critical in establishing breastfeeding. The women identified aspects of clinical breastfeeding support during their hospital stay that negatively impacted breastfeeding establishment, such as a lack of explanation during physical help, conflicting information, and rushed care. The delayed initiation of breastfeeding, higher pain scores, and a negative birth experience were more prevalent after an NELUSCS birth, while primiparous women were more likely to have delayed breastfeeding initiation. Reports of commercial milk formula use were significantly higher in primiparous women and in those in a private hospital setting (Table 5). Women that anticipated some challenges when learning to breastfeed reported the ability to maintain confidence and persist when facing breastfeeding difficulties.

The >70% prevalence of early breastfeeding initiation after a CS birth was compared favourably with the estimated early initiation rate of 51.4% for upper-middle-income countries [23]; however, these rates were 50% lower after an NELUSCS birth and 30% lower for primiparous women (Table 3). Primiparous women accounted for >75% of the NELUSCS group, and the observed differences based on parity and CS birth type are supported by reported positive associations between multiparity and early breastfeeding initiation [24]. Furthermore, the known barriers to early initiation, including birth complications and post-operative pain, were more prevalent after an NELUSCS birth [25]. Given the strong positive associations with breastfeeding exclusivity, both early breastfeeding initiation and skin-to-skin contact should be prioritised wherever possible after an NELUSCS birth. [24].

The reported early post-operative pain levels were higher after an NELUSCS birth, with progression to severe pain being more prevalent (Table 2 and Table 3). This may be explained, in part, by the predominant use of epidural anaesthesia for NELUSCS births, which is associated with higher ratings of pain when compared to spinal anaesthesia that is commonly used for ELUSCS births [26]. Inadequate postpartum pain management has been shown to negatively impact breastfeeding initiation and exclusivity during hospital stays [9,27], and it is associated with chronic pain at three months postpartum [28]. Both the modification of existing strategies and the use of non-pharmacological pain management strategies merit further investigation. A recent Scottish study reported that the maternal self-administration of oral analgesia during postpartum hospital stay was associated with significantly higher pain satisfaction compared to when oral analgesia was administered by midwives [29]. As the study did not compare outcomes based on birth mode, an investigation of the effectiveness of self-administration following elective and non-elective CS births is needed. A review of complementary and alternative therapies for post-CS pain found a low level of evidence to support the concurrent use of prescribed analgesia and aromatherapy, music therapy, or electromagnetic therapy in reducing pain in the first 24 h after a CS birth [30]. Although it is recognized that psychosocial factors influence the experience of pain, few studies have examined these factors in relation to postpartum pain. A large Brazilian study found that pre-operative anxiety was predictive of moderate to severe acute post-CS pain [31], suggesting that partner engagement and support may improve maternal wellbeing outcomes.

While >70% women reported moderate to severe pain during the early postpartum period, <25% reported that pain impacted their breastfeeding experiences. A third of the women highlighted reduced mobility as a limiting factor for breastfeeding and infant care, and this was a major theme in the qualitative data. This is similar to the findings from studies in the UK, South Africa, and Australia where women identified the effects of postoperative pain, such as impaired mobility and fatigue, as contributors to breastfeeding difficulties after a CS birth [24]. The optimisation of pain management and functional recovery is needed after NELUSCS birth, with more consistent application of published stepwise multimodal analgesia guidelines [32].

Impaired mobility resulted in women’s dependence on others to pass them the baby for feeding. The identified complicating factors, such as the interference of medical devices, separation from their babies, and the absence of clinical staff or their partners to provide physical assistance, were consistent with the published evidence [33,34]. However, the construction of the ‘reduced mobility’ subtheme revealed a major disparity according to the hospital setting. Women who birthed in a private hospital could have their partners stay with them overnight and identified their partners’ physical and emotional support as fundamental to breastfeeding establishment. In a public hospital setting, outside of ‘visiting hours’, women were reliant on clinical staff who were often too busy, leaving them all alone. This finding correlates with reports of inadequate clinical support as a barrier to establishing breastfeeding after a CS birth [34,35]. Given the barriers posed by reduced mobility, strategies are required to assist women during their hospital stays. The facilitation of partner support and more flexible visiting arrangements in public hospitals have the potential to improve breastfeeding outcomes [36]. While there are scant published data on maternity units’ designated visiting hours for partners, there is global evidence of maternal distress and isolation as a result of extremely limited or banned visiting during the COVID-19 pandemic [37]. Evidence from a variety of socio-cultural settings indicates that most partners value and want to support breastfeeding [38,39,40]. Where partners cannot be available to assist, the use of sidecar bassinets has facilitated breastfeeding through ease of infant access and was strongly preferred over free-standing bassinets by women after an ELUSCS birth [41]. The use of side-car bassinets in maternity units warrants further evaluation.

Less than 60% of women were satisfied with the breastfeeding help received in hospital (Table 4). The lower satisfaction rating and identified clinical care barriers to breastfeeding, including conflicting advice and ‘doing not showing’, are consistent with those of previous Australian studies [34,42] as well as those from studies in other high-income countries [35,43]. Another subtheme of ‘too busy to help’ corroborates with a recent study where women’s satisfaction with postnatal care was positively associated with staffing levels [44], likely reflecting the current midwifery staffing crisis. Overall, care was described as helpful when it was unrushed, consistent, and personalised, and was characterised as unhelpful when it was hurried and lacked explanations. A recent Cochrane review found that, across a range of settings, dedicated early breastfeeding support is associated with higher exclusivity and duration rates, with possible sources of support including midwives, international-board-certified lactation consultants, and peer supporters [45]. Women’s experiences of establishing breastfeeding can be optimised through the application of adult learning principles whereby midwives partner with women to determine their attitudes, perceptions, and learning needs; show and explain rather than do; and provide consistent and constructive feedback [46]. It is imperative that midwives are afforded the time needed to provide adequate breastfeeding care and support. Healthcare systems need to support adequate staffing and more flexible visiting for partners and implement evidence-based breastfeeding programs, such as the Baby-Friendly Initiative, to ensure appropriate breastfeeding support. The Baby-Friendly Initiative achieves greater consistency of care with global evidence of higher rates of maternal satisfaction as well as higher breastfeeding exclusivity and duration rates [47,48].

The risks of reporting the birth experience as ‘a little traumatic’ or ‘very traumatic’ and unmet birth expectations was ten times higher after an NELUSCS birth when compared to an ELUSCS birth. This is congruent with the findings of a Greek study where a lack of support and early breastfeeding cessation also contributed to post-traumatic stress [49], and is likely associated with fear and a lack of control that are more commonly experienced with NELUSCS births [50]. It is important that women are screened and offered support for low birth satisfaction, as this is associated with higher risks of postnatal depression and post-traumatic stress disorder [51,52], as well as lower breastfeeding exclusivity and duration rates [53].

The mixed-methods study design, large sample size, and diversity of CS birth circumstances represent the key strengths of this study. However, the experiences of women that had a CS birth due to immediate life-saving measures were not differentiated from less emergent NELUSCS births. Also, while the use of social media for recruitment to health research results in a more representative sample [54], there was an under-representation of women with socio-economic disadvantages. Furthermore, women who birthed in public hospitals were under-represented, accounting for 50% of the study sample and 75% of all Australian births [55]. This study may be limited by self-selection bias, as most participants identified as Australian, had completed tertiary education, and had a longer intended breastfeeding duration. Therefore, the findings may not accurately reflect the experiences of culturally and linguistically diverse and disadvantaged populations within the Australian society. Consumer engagement with under-represented groups will be critical for the design of future studies. The adequate capture of under-represented women’s early breastfeeding experiences and the facilitators and barriers can inform their maternity care. Overall, this study’s findings provide unique insights into the strengths and areas that need urgent improvement in clinical care.

## 5. Conclusions

Women face challenges in establishing breastfeeding after a CS birth, including the physical challenges and lack of help to pick up the baby for breastfeeding, conflicting advice, and rushed care during their hospital stays. These difficulties are exacerbated after an NELUSCS birth due to the delayed initiation of breastfeeding and higher ratings of pain and psychosocial distress, while women that birth in public hospitals struggle with reduced access to partner support. Women reported benefiting from midwives that took the time to show and explain the process when helping with breastfeeding, and women who anticipated that learning to breastfeed would involve some trial and error found the process less stressful.

Maternity care providers can influence breastfeeding at the individual level. However, adequate staffing in postnatal wards with time to implement adult learning principles and evidence-based guidelines is needed to foster consistency in clinical lactation support. Health service policies and practices, such as open visiting for partners and the use of side-car bassinets, may increase support and enable women to access their infants more easily for breastfeeding. Lastly, governments can enable equitable and consistent breastfeeding support by legislating, monitoring, and financing healthcare policies to ensure sustainability. Nation-wide scaling up and support of the Baby-Friendly Initiative offers a cost-effective strategy that increases early breastfeeding initiation, exclusivity, and duration to improve breastfeeding outcomes after CS births and subsequent infant and maternal health outcomes.

## Figures and Tables

**Figure 1 ijerph-21-00296-f001:**
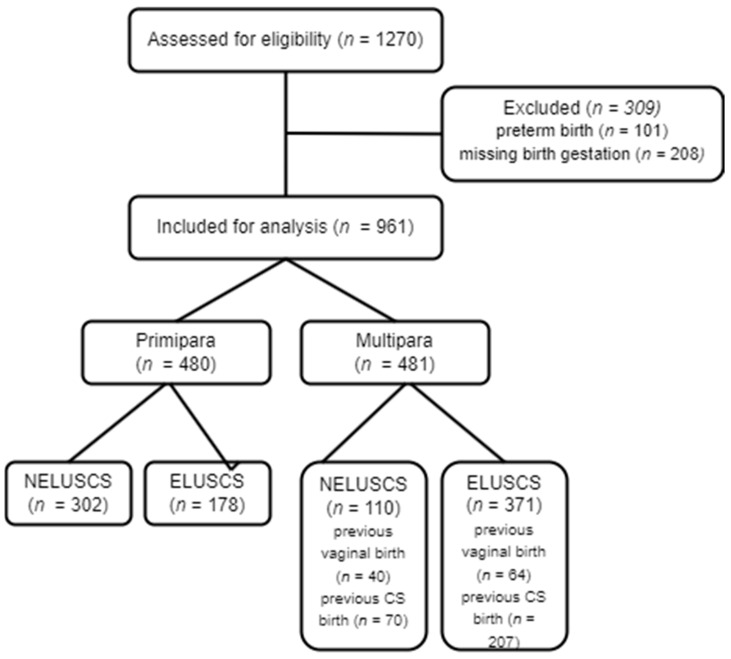
Study recruitment flow chart.

**Figure 2 ijerph-21-00296-f002:**
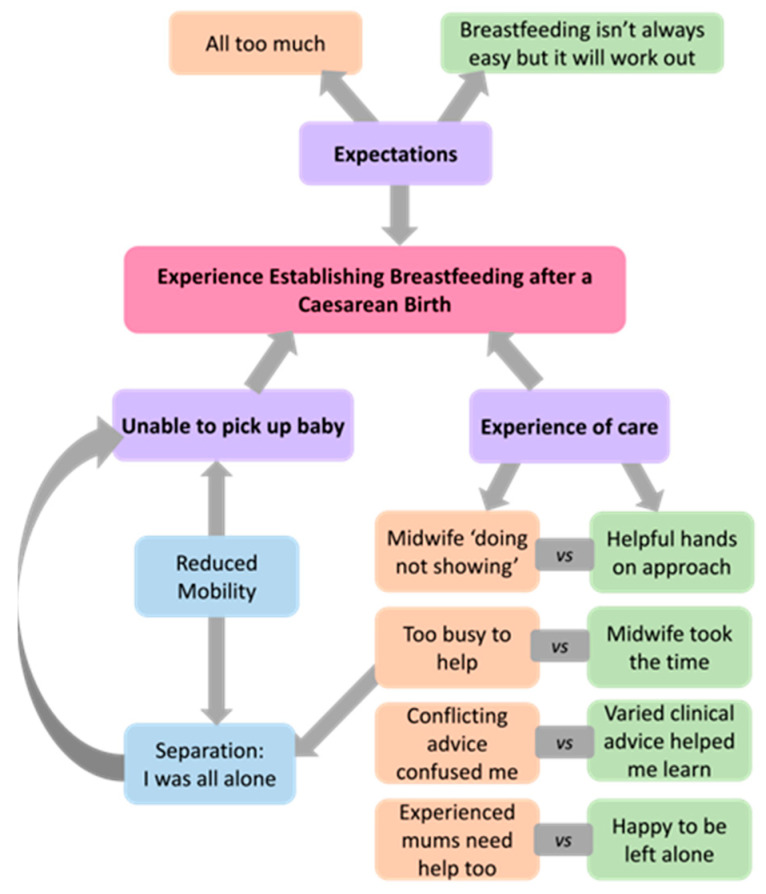
Women’s experiences of establishing breastfeeding after a caesarean birth.

**Table 1 ijerph-21-00296-t001:** Participants’ sociodemographic characteristics.

	Overall (*n* = 961)	NELUSCS (*n* = 412)	ELUSCS (*n* = 549)
BMI category			
Underweight (<18.5)	17 (1.8)	5 (1.2)	12 (2.2)
Normal (18.5–24.9)	435 (45.3)	182 (44.2)	253 (46.1)
Overweight (25.0–29.9)	258 (26.8)	115 (27.9)	143 (26.0)
Obese (≥30.0)	241 (25.1)	105 (25.5)	136 (24.8)
Missing	10 (1.0)	5 (1.2)	5 (0.9)
SEIFA index percentile			
0–30 most disadvantaged	96 (10.0)	41 (10.0)	55 (10.0)
30–70	415 (43.2)	193 (46.8)	222 (40.4)
70–100 least disadvantaged	444 (46.2)	177 (43.0)	267 (48.6)
Missing	6 (0.6)	1 (0.2)	5 (0.9)
Ethnic group			
Australian	722 (75.1)	307 (74.5)	415 (75.6)
British	108 (11.2)	49 (11.9)	59 (10.7)
Aboriginal or TSI	23 (2.4)	14 (3.4)	9 (1.6)
Other	217 (22.6)	98 (23.8)	430 (78.3)

Data reported as mean ± SD or *n* (%); 1 = sum of percentages > 100%, as women could select multiple responses. BMI = body mass index, ELUSCS = elective lower uterine segment caesarean section, NELUSCS = non-elective lower uterine segment caesarean section, SD = standard deviation, SEIFA = Socio-Economic Indexes for Australia (categorises postal area codes by relative socio-economic advantage and disadvantage), TSI = Torres Strait Islander.

**Table 2 ijerph-21-00296-t002:** Birth experience and breastfeeding initiation based on caesarean birth type and parity.

	Overall	NELUSCS	ELUSCS	Primipara	Multipara
	(*n* = 961)	(*n* = 412)	(*n* = 549)	(*n* = 480)	(*n* = 481)
Birth at public hospital	493 (51.3)	263 (63.8)	230 (41.9)	237 (49.4)	256 (53.2)
Caesarean anaesthesia					
Spinal block	517 (53.8)	99 (24.0)	418 (76.1)	194 (40.4)	323 (67.2)
Epidural	221 (23.0)	177 (43.0)	44 (8.0)	152 (31.7)	69 (14.3)
Combined	148 (15.4)	88 (21.4)	60 (10.9)	99 (20.6)	49 (10.2)
General anaesthetic	32 (3.3)	28 (6.8)	4 (0.7)	15 (3.1)	17 (3.5)
Unsure/missing	43 (4.4)	20 (4.8)	23 (4.2)	20 (4.1)	23 (4.7)
Pain rating after birth	6.0 (4.0, 8.0)	7.0 (5.0, 8.0)	6.0 (4.0, 7.0)	6.0 (5.0, 8.0)	6.0 (4.0, 7.75)
Birth complications					
None	700 (72.8)	257 (62.4)	443 (80.7)	335 (69.8)	365 (75.9)
Infant resuscitation	33 (3.4)	24 (5.8)	9 (1.6)	19 (4.0)	14 (2.9)
Neonatal unit admission	117 (12.2)	63 (15.3)	54 (9.8)	49 (10.2)	68 (14.2)
Postpartum haemorrhage	68 (7.1)	41 (10.0)	27 (4.9)	31 (6.5)	37 (7.7)
Birth experience rating					
Quite easy	444 (46.2)	58 (14.1)	386 (70.3)	172 (35.8)	272 (56.5)
Difficult but overall okay	221 (23.0)	127 (30.8)	94 (17.1)	120 (25.0)	101 (21.0)
A little traumatic	197 (20.5)	140 (34.0)	57 (10.4)	123 (25.6)	74 (15.4)
Very traumatic	99 (10.3)	87 (21.1)	12 (2.2)	65 (13.5)	34 (7.1)
Birth expectations met	545 (56.7)	93 (22.6)	452 (82.3)	211 (44.0)	334 (69.4)
BF initiated ≤ 1 h birth	675 (70.2)	245 (59.5)	430 (78.3)	304 (63.4)	371 (77.1)

Reported as n (%) or median (Q1 and Q3). ELUSCS = elective lower uterine segment caesarean section; NELUSCS = non-elective lower uterine segment caesarean section.

**Table 3 ijerph-21-00296-t003:** Univariable and multivariable logistic regression analysis of factors associated with early initiation of breastfeeding.

Covariate	Univariate Analysis Odds Ratio (95% CI)	*p*	Multivariate Analysis Odds Ratio (95% CI)	*p*
(Intercept)			5.89 (3.65–9.02)	<0.001
CS type: NELUSCS	0.40 (0.30–0.53)	<0.001	0.50 (0.36–0.68)	<0.001
Parity: primipara	0.50 (0.37–0.66)	<0.001	0.68 (0.50–0.94)	0.02
Pain score after birth	0.92 (0.86–0.98)	0.011	0.95 (0.89–1.02)	0.13

CS = caesarean section birth, NELUSCS = non-elective lower uterine segment caesarean section.

**Table 4 ijerph-21-00296-t004:** Maternal ratings of pain and infant feeding outcomes during postpartum hospital stay.

	Overall (*n* = 961)	NELUSCS (*n* = 412)	ELUSCS (*n* = 549)	Primipara (*n* = 480)	Multipara (*n* = 481)
Pain rating					
Mild (≤3)	158 (16.4)	42 (10.2)	116 (21.1)	62 (12.9)	96 (20.0)
Moderate (4–6)	351 (36.5)	146 (35.4)	205 (37.3)	176 (36.7)	175 (36.4)
Severe (≥7)	426 (44.3)	210 (51.0)	216 (39.3)	231 (48.1)	195 (40.5)
Missing	26 (2.7)	14 (3.4)	12 (2.2)	11 (2.3)	15 (3.1)
Pain impacted BF					
Yes	197 (20.5)	129 (31.3)	68 (12.4)	125 (26.0)	72 (15.0)
No	689 (71.7)	247 (60.0)	442 (80.5)	323 (67.3)	366 (76.1)
Missing	75 (7.8)	36 (8.7)	39 (7.1)	32 (6.7)	43 (8.9)
Feeding method					
Breastfeeding	771 (80.2)	311 (75.5)	460 (83.8)	356 (74.2)	415 (86.3)
Expressed milk	230 (23.9)	113 (27.4)	101 (21.0)	129 (26.9)	101 (21.0)
Infant formula	308 (32.1)	154 (37.4)	108 (22.5)	200 (41.7)	108 (22.5)
Hospital stay (days)	4 (3,5)	4 (3,5)	4 (3,5)	4 (3,5)	4 (3,5)
Missing	26 (2.7%)	12 (3.4%)	12 (2.2%)	11 (2.3%)	15 (3.1%)

Reported as n (%) or median (Q1 and Q3). BF = breastfeeding, ELUSCS = elective lower uterine segment caesarean section, NELUSCS = non-elective lower uterine segment caesarean section.

**Table 5 ijerph-21-00296-t005:** Univariable and multivariable logistic regression analyses of factors associated with infant feeding of commercial milk formula during postpartum hospital stay.

Covariate	Univariate Analysis Odds Ratio (95% CI)	*p*	Multivariate Analysis Odds ratio (95% CI)	*p*
(Intercept)	0.37 (0.25–0.56)	<0.001		
CS type: NELUSCS	1.40 (1.07–1.82)	0.013	1.02 (0.71–1.47)	0.91
Parity: primipara	2.23 (1.71–2.92)	<0.001	2.16 (1.60–2.91)	<0.001
Pain score after birth	1.07 (1.01–1.13)	0.033	1.05 (0.99–1.12)	0.12
Birth at private hospital	1.52 (1.17–1.98)	0.002	1.67 (1.25–2.32)	0.001
Birth complications	2.16 (1.61–2.90)	<0.001	2.25 (1.64–3.10)	<0.001
Unmet birth expectations	1.26 (0.96–1.63)	0.091	0.91 (0.63–1.31)	0.61

CS = caesarean section birth, NELUSCS = non-elective lower uterine segment caesarean section.

**Table 6 ijerph-21-00296-t006:** Univariable and multivariable logistic regression analyses of factors associated with a negative caesarean birth experience.

Covariate	Univariate Analysis Odds Ratio (95% CI)	*p*	Multivariate Analysis Odds Ratio (95% CI)	*p*
(Intercept)			0.02 (0.01–0.04)	<0.001
CS type: NELUSCS	11.58 (8.21–16.60)	<0.001	10.21 (6.88–15.43)	<0.001
Parity: primipara	2.34 (1.75–3.14)	<0.001	0.98 (0.66–1.44)	0.92
Pain score after birth	1.29 (1.20–1.39)	<0.001	1.23 (1.13–1.34)	0.13
Birth complications	4.80 (3.53–6.55)	<0.001	3.95 (2.74–5.72)	<0.001

CS = caesarean section birth, NELUSCS = non-elective lower uterine segment caesarean section.

## Data Availability

The data presented in this study are available upon request from the corresponding author, S.L.P. The data are not publicly available due to privacy restrictions.

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
