# Peer review of "Australian Women’s Experiences of Establishing Breastfeeding after Caesarean Birth"

_ijerph, 2024, doi:10.3390/ijerph21030296_

Round 1

Reviewer 1 Report

Comments and Suggestions for Authors

- To add to the introduction information about the average length of breastfeeding in general
- Add a characterization of the online questionnaire (created by the authors or adopted or standardized...?)
- Add to the selection of respondents information on exclusion or inclusion of multiple gestation (twins...), completion of childbirth preparation/other training on newborn/infant care
- In the qualitative analysis, reduce or omit Table 6 and the text related to the table (not directly related to breastfeeding, but related to factors related to CS)
- Qualitative analysis - in the individual themes, add the ratio of primipara vs. multipara statements (several statements are linked to primipara)
- In the title of theme III - add also the subtheme of reduced mobility
- Add information on average length of hospital stay after CS, choice of private hospital (in relation to SEIFA index), and presence of a lactation consultant on site (or is every PA also a lactation consultant?) to the discussion

Reviewer 2 Report

Comments and Suggestions for Authors

Dear authors,

I found this work very interesting, which addresses the experiences, facilitating factors and barriers of Australian women in establishing breastfeeding after a cesarean birth.

Before completing the review that is being carried out, it is essential to correct an important error related to the sample expressed in figure 1, which indicates that MULTIPARA n= 481: NELUSCS n=110 and ELUSCS n=271; so 110 + 271=381 and not 481 as indicated in the work, that is, 100 less than what is being indicated; If so, the total sample n= 961 would also be erroneous, as it was actually 861.

Likewise, in table 1, ELUSCS n=549 would be wrong, since according to figure 1 it would have to be n= 449. The same thing happens in tables 2, 4...

It is urgent that these numerical data be reviewed, both “n” (MULTIPARA) and n=961 in all statistical analyses, tables, results and discussion.

All the best.

Reviewer 3 Report

Comments and Suggestions for Authors

A well-written paper addressing a significant area of research with clarity.

Few minor suggestions would be,

1. Need further details to elucidate the uniqueness of this study. Specifically, integrating a brief discussion on prior research examining facilitators and barriers to breastfeeding within similar demographic contexts would provide valuable context.

2. Additionally, insight into the authors' intentions regarding future studies, particularly pertaining to women experiencing various levels of disadvantage, would further enrich the paper's relevance and potential implications.

Reviewer 4 Report

Comments and Suggestions for Authors

The manuscript adds valuable insights into improving breastfeeding outcomes and clinical care for women after cesarean section births. Further comparison with international research could enhance understanding of how contextual factors influence breastfeeding after CS. Expanding the discussion on policy implications within healthcare could strengthen the impact of the study’s recommendations.

Abstract

Line 15-16: Consider briefly describing the study’s design, and methods of data collection and analysis, allowing readers to quickly understand how the research was conducted.

Line 24: Consider outlining the study’s conclusions.

Introduction

Line 49: Change “re-ported” to “reported”.

Line 51: Consider identifying the gaps based on previous research in this field.

Line 53: Change “re-search” to “research”.

Materials and methods

Line 86: Consider providing detailed information on the development and validation of the questionnaire. Information on the development and validation of the questionnaire. How the questionnaire items were developed, and tested for reliability and validity.

Line 94: While the paper mentioned the use of inductive thematic analysis for qualitative data, more detailed information on how themes were derived and validated could help readers understand the process.

Discussion:

Consider adding a more extensive exploration of international studies to compare and contrast the findings in a global context.

The discussion could benefit from a deeper discussion about specific policy implications (e.g., how healthcare systems can structurally support the recommended changes in clinical practice to improve breastfeeding support post-CS.

Line 421: while the discussion mentions pain as a barrier to breastfeeding, a more detailed exploration of effective pain management strategies could be beneficial (e.g., discussing non-pharmacological methods and how they can be integrated into post-CS care)

Round 2

Reviewer 2 Report

Comments and Suggestions for Authors

Dear authors,

These investigations are very necessary to reinforce breastfeeding and prevent its early abandonment.

You have done a good job

Greetings.

Reviewer 4 Report

Comments and Suggestions for Authors

The authors addressed all of my comments well.